# The Negative Impact of Sarcopenia on Hepatocellular Carcinoma Treatment Outcomes

**DOI:** 10.3390/cancers16132315

**Published:** 2024-06-24

**Authors:** Annalisa Cespiati, Daniel Smith, Rosa Lombardi, Anna Ludovica Fracanzani

**Affiliations:** 1SC Medicina ad Indirizzo Metabolico, Fondazione IRCCS Cà Granda Ospedale Maggiore Policlinico, via F. Sforza 35, 20122 Milan, Italy; daniel.smith@unimi.it (D.S.); rosa.lombardi@unimi.it (R.L.); anna.fracanzani@unimi.it (A.L.F.); 2Department of Pathophysiology and Transplantation, University of Milan, 20122 Milan, Italy

**Keywords:** hepatocellular carcinoma, sarcopenia, HCC treatment, liver transplantation, resection, locoregional therapies, systemic therapies, overall survival, HCC recurrence

## Abstract

**Simple Summary:**

This paper discusses the importance of addressing sarcopenia in patients with hepatocellular carcinoma (HCC). The aim is to analyze how sarcopenia affects treatment outcomes for HCC, including liver transplantation, surgical resection, locoregional treatments, and systemic therapies. Sarcopenia is prevalent among HCC patients and independently correlates with lower overall survival, recurrence-free survival, and progression-free survival across all treatment modalities. Sarcopenia also increases the rate and severity of adverse events, particularly in surgery and systemic therapies. This research highlights the need for evaluating sarcopenia before HCC treatment initiation to better predict patient prognosis and tailor treatment approaches accordingly. However, the impact of sarcopenia on HCC recurrence and spread beyond the liver remains poorly understood, indicating a need for further research in this area. Overall, this research sheds light on the significance of considering sarcopenia in HCC management and may prompt efforts to identify therapies that can address muscle loss in these patients, potentially improving treatment outcomes and patient care.

**Abstract:**

Introduction: Hepatocellular carcinoma (HCC) represents a major global health concern, characterized by evolving etiological patterns and a range of treatment options. Among various prognostic factors, sarcopenia, characterized by loss of skeletal muscle mass, strength, and function, has emerged as a pivotal contributor to HCC outcomes. Focusing on liver transplantation, surgical resection, locoregional treatments, and systemic therapies, this review aims to analyze the impact of sarcopenia on HCC treatment outcomes, shedding light on an underexplored subject in the pursuit of more personalized management. Methods: A comprehensive literature review was conducted by searching peer-reviewed articles on sarcopenia and treatment outcomes in patients with HCC from inception up to October 2023. Results: Sarcopenia was found to be prevalent among HCC patients, exhibiting different occurrence, possibly attributable to diverse diagnostic criteria. Notably, despite variations in studies utilizing skeletal muscle indices, sarcopenia independently correlated with lower overall survival (OS), recurrence-free survival (RFS), and progression-free survival (PFS) across surgical (both transplantation and resection), locoregional, and systemic therapies, including tyrosine-kinase inhibitors (TKIs) and immune-checkpoint inhibitors (ICIs). Moreover, a link between sarcopenia and increased rate and severity of adverse events, particularly in surgery and TKIs recipients, and larger tumor size at diagnosis was observed. While baseline sarcopenia negatively influenced treatment outcomes, alterations in muscle mass post-treatment emerged as primary determinants of reduced OS. Conclusions: Sarcopenia, either present before or after HCC treatment, negatively correlates with response to it, across all etiologies and therapeutic strategies. Although only a few studies have evaluated the impact of supervised physical activity training on muscle mass and OS after HCC treatment, it is crucial to evaluate the presence of sarcopenia before treatment initiation, to better stratify patients’ prognosis, thus performing a more tailored approach, and identify therapies able to restore muscle mass in HCC patients. Conversely, the impact of sarcopenia on HCC recurrence and extrahepatic spread remains inadequately explored.

## 1. Introduction

Hepatocellular carcinoma (HCC) is the fourth-leading cause of cancer-related mortality worldwide, and it is the most common form of primary liver cancer, typically arising alongside chronic liver disease (CLD) [1]. Over the last few years, the etiological landscape of HCC has dramatically changed with a decline in cases attributed to viral hepatitis (hepatitis B virus, HBV, and hepatitis C virus, HCV) and a rise in those associated with metabolic dysfunction associated liver disease (MASLD) and alcohol [2,3].

Survival rates in HCC are extremely variable and depend on several factors like age, ethnicity, gender, access to screening programs, and treatment decisions [4]. Additionally, the severity of the underlying CLD has emerged as a major determinant of prognosis for patients with HCC. Factors such as a low-performance status, Child-Pugh classes B and C, the presence of portal vein thrombosis, and esophageal varices are independently associated with poor survival rates [5]. In addition, disparities in HCC survival rates are attributed to economic barriers and a lack of awareness regarding screening programs for at-risk patients [6].

Several treatment options have been proposed for HCC, some of them aimed at curing the disease, and others at controlling it. Among therapies with curative intent, liver transplantation and surgical resection offer higher overall and progression-free survival (PFS) rates [7], followed by locoregional treatment, such as ablation [8]. Other locoregional therapies like transarterial chemoembolization (TACE) and transarterial radioembolization (TARE), defined also as selective internal radiation therapy (SIRT), provide local control of HCC and are valuable for downstaging tumors [9,10]. Systemic therapies are recommended for advanced tumor diseases or for those not suitable for other treatments [9]. Along with the widely used tyrosine kinase inhibitor (TKI), the most known namely sorafenib, in 2020 immunotherapies became available for HCC, showing higher overall survival (OS) and PFS compared to TKI [11].

Treatment choices are influenced by the characteristics of patients and the neoplastic disease, according to the Barcelona Clinic Liver Cancer (BCLC) algorithm [9]. The 2022 BCLC strategy considers the characteristics of the tumor, such as number and dimension of nodules, portal invasion, and extrahepatic spread, along with liver function and patients’ physical status related to neoplasia [9]. Nevertheless, several comorbidities impact both on OS and treatment response. For example, type 2 diabetes mellitus (T2DM) and obesity promote HCC development and negatively impact the response to treatments, possibly due to systemic inflammation and immune modulations [12].

In addition to these factors, sarcopenia is emerging as a pivotal factor in the development of HCC [13]. Sarcopenia, characterized by the loss of skeletal muscle mass, quality, and strength [14], is categorized as primary when associated with age and secondary when linked to other chronic diseases [15]. Within CLD, sarcopenia is highly prevalent at all stages, exhibiting a higher prevalence in advanced CLD conditions like cirrhosis, HCC, and the pre-transplant setting [16]. Various imaging techniques such as dual energy X-ray absorptiometry (DXA), computed tomography (CT), magnetic resonance imaging (MRI), or bioimpedance analysis (BIA) assess muscle mass, while physical tests evaluate muscle quality and strength [14].

Sarcopenia is associated with several neoplastic diseases, including HCC, with a pooled prevalence of 39%. However, the prevalence of sarcopenia in HCC ranges widely from 11 to 85%, potentially due to different methods and cut-offs used to define sarcopenia and the heterogeneous population included in the studies [17]. Systemic inflammation, malnutrition, and physical inactivity are the features shared by sarcopenia and HCC [18]. The presence of sarcopenia, along with visceral adiposity, adversely impact the prognosis of HCC patients [17,19]; in fact, individuals with HCC and sarcopenia experience reduced OS, increased rates of tumor recurrence after surgery, and increased post-treatment adverse effects, particularly in the early stages of HCC [20]. Furthermore, the coexistence of sarcopenia and increased body mass index (BMI), a condition named sarcopenic obesity, further reduces the OS in HCC patients [21].

Lastly, myosteatosis, defined as the infiltration of adipose tissue into muscles, has been associated with several malignancies [22], nevertheless its role in HCC is undetermined [23].

While the detrimental impact of sarcopenia on HCC development, OS, and poor prognosis is well recognized, a comprehensive understanding of its influence on HCC treatment outcomes remains to be fully elucidated.

This narrative review aims to assess the impact of sarcopenia on treatment response and tumor recurrence following liver transplantation, surgical resection, locoregional treatment, and systemic therapies in HCC patients.

## 2. Research Strategy and Study Selection

We conducted a literature review by searching peer-reviewed articles about sarcopenia and treatment response in patients with HCC on the PubMed Database from inception up to October 2023. The search strategy combined terms related to sarcopenia and HCC treatment outcomes in patients who underwent liver transplantation, resection, locoregional treatment, and systemic therapy. We included experimental and observational studies, case reports, clinical trials, editorials, and commentaries that analyzed, as a primary or secondary outcome, the impact of sarcopenia on OS, PFS, and adverse events in patients with HCC undergoing specific treatments, such as liver transplantation, resection, locoregional, or systemic therapy. We excluded studies that did not meet the selection criteria, abstracts, and articles not written in English. We also excluded studies that focused solely on the impact of sarcopenia on the development of HCC without assessing its effect on treatment response. A manual search was performed by evaluating the references of included studies. AC and DS reviewed the articles independently and any discrepancies were resolved by RL.

The literature search provided a total of 140 articles. A significant portion of them was excluded as their outcomes did not align with the aim of our review. Eighty-five studies met the inclusion criteria and were included in this review.

Table 1 summarizes the parameters used to define sarcopenia and myosteatosis across the studies included in the review.

## 3. Impact of Sarcopenia on Surgical Option Outcomes

Tumor resection and orthotopic liver transplantation (OLT) are considered the interventions for HCC with the highest curative potential. Surgical resection is typically reserved for patients with a single HCC nodule and without signs of clinical significant portal hypertension (CSPH). On the other hand, OLT is recommended for patients in the early stages of HCC (0–A) within Milan criteria [24], but it may be considered for patients at the intermediate stage (B) if they meet the extended liver transplant criteria specific to each transplant center [9].

The impact of sarcopenia in the context of OLT for HCC has been specifically investigated only in five retrospective studies, while 28 studies have explored the role of sarcopenia on surgical resection outcomes, with two of these being prospective.

Appendix A summarizes the general characteristics of studies evaluating the impact of sarcopenia on liver transplant and surgical resection outcomes.

### 3.1. Impact of Sarcopenia on Liver Transplantation

In addition to the Milan criteria, several extended liver transplant criteria have been developed, such as up-to-seven criteria (the sum of the largest tumor diameter and the number of tumor nodules should not exceed seven) [25], and the University of California San Francisco (UCSF) criteria (unique tumor nodule smaller than 6.5 cm or up to three tumors with the largest smaller than 4.5 cm, and the sum of the diameters of the three nodules being less than 8 cm) [26]. The 5-year OS after OLT is comparable between patients meeting Milan criteria and those meeting extended liver transplant criteria, ranging from 65% to 78% [26,27]. For patients who are not initially eligible for OLT but undergoing effective downstaging, the 10-year OS is approximately 52%, indicating that OLT after downstaging is a viable treatment option for HCC patients [28].

The presence of sarcopenia prior to OLT, expressed as a reduction in skeletal muscle index (SMI) and psoas muscle mass index (PMI), a valid marker of skeletal muscle mass calculated as the sum of the left and right psoas major muscle area at CT scan divided by height squared [29], is independently associated with a decrease in 5-year OS after transplant [30,31], particularly among males [31]. Beyond muscle mass, body composition is a crucial predictor of OS. In fact, patients with a low skeletal muscle mass to visceral fat area (SVR) witnessed a reduction in OS after OLT [32]. In only one study, sarcopenia did not show a statistically significant association with OS [33].

The impact of sarcopenia on recurrence-free survival (RFS) after OLT remains a topic of debate. Despite the study conducted by Tan and colleagues observing a negative correlation between sarcopenia and low RFS, the association did not achieve statistical significance [31]. Conversely, Itoh et al. showed that patients with a low SVR had a lower RFS [32].

Sarcopenia also negatively affects the postoperative period, with a threefold higher risk of complications such as the development of biloma, infections, liver insufficiency, renal failure, cardiac arrest, and hemorrhage [33], leading to an increased hospital stay [34].

### 3.2. Impact of Sarcopenia on Surgical Resection

Surgical resection is mainly indicated in the early stages (0–A) of HCC according to BCLC guidelines [9]. The primary objective of surgical resection is to excise the tumor along with an adequate margin of healthy liver tissue, aiming at eradicating the cancerous lesion entirely. This curative approach is particularly applicable when the tumor burden is localized and the vascular invasion or the extrahepatic spread is absent or minimal.

Among the 28 studies examining surgical resection, 16 evaluated the effect of sarcopenia on OS and RFS across all HCC etiologies. The majority of the studies concluded that sarcopenia, defined as the reduction in both SMI and PMI, played a major role in negatively affecting HCC prognosis in terms of OS and RFS (by decreasing it by about 30–50%) [35,36,37,38,39,40,41,42,43]. Conversely, in two studies [44,45] with limited cohorts (100 and 157 HCC patients from various etiologies), no statistically significant differences were found regarding OS and RFS in sarcopenic patients compared to non-sarcopenic. Furthermore, one study showed discordant results concerning OS [33]. In fact, Valero and colleagues focused on post-operative complications after resection and transplantation in both patients with HCC and intrahepatic cholangiocarcinoma. While they found a significant correlation between sarcopenia and post-operative complications, there was no correlation with long-term outcomes, likely due to the differing long-term prognoses of these two diseases, as noted by the authors. Regarding RFS, only one study found no significant correlation with sarcopenia [46].

Five studies [33,47,48,49,50] evaluated the impact of sarcopenia on major post-operative complications. Again, in all studies, sarcopenia was related to worse outcomes, in terms of increased in-hospital stay and hospitalization costs, rates of occurrence and severity of major complications (Clavien-Dindo classification of grade 3 or 4) requiring surgical or radiological intervention, and life-threatening organ failure and death.

Interestingly, in a Japanese retrospective study [51] sarcopenia was demonstrated to predict increased short- and long-term post-operatory complications and worse OS even independently of the pre-operatory risk as determined by the American Society of Anesthesiologists (ASA) score.

Finally, one multicenter retrospective cohort study [52] evaluated the impact of sarcopenia and its predictive performance after liver resection in 553 HCC patients of different etiology, divided into two cohorts, one from the Netherlands and the other one from Japan. This study resulted in a significant negative association between sarcopenia and OS in the Japanese cohort, without the same evidence in the Dutch one. These results suggest that there might be a different impact of sarcopenia depending on ethnicity.

As said, sarcopenia and obesity, when coexisting, cooperate in worsening the outcomes of patients with HCC, as well as the patients’ nutritional status.

This has been shown by two Asian retrospective studies [53,54], evaluating more than 400 patients (mostly male and with viral etiology), that demonstrated that sarcopenic obesity was an independent risk factor for mortality and HCC recurrence, whereas both sarcopenia and obesity (as defined by BMI) did not impact the OS and RFS when considered alone.

Another study from Japan retrospectively evaluated the impact of sarcopenia and obesity on the OS, RFS, and post-operative complications of HCC patients after resection, by using the visceral fat area (VFA) as an estimation of obesity instead of BMI, assessed by CT measurement at the level of the umbilicus. However, their results showed that only sarcopenia negatively afflicted these outcomes and that VFA inversely correlated with them, possibly because patients with high VFA had more muscle mass and better nutritional status [46].

Regarding myosteatosis, three retrospective studies [44,55,56] have shown its association with an increased risk of post-operative complications and worse OS and RFS. In particular, two of them [55,56] highlighted how, after resection, changes in muscle quality, as determined by CT evaluation of intramuscular adipose tissue content (IMAC) rather than quantity, as determined by the psoas muscle area at CT scan, predicted worse OS and RFS. This discrepancy may be possibly due to the fact that PMI measurement includes both muscle and intramuscular adipose tissue (IMAT), making PMI a less reliable indicator of sarcopenia.

Lastly, an Asian study following up 303 patients affected by HCC from different etiologies (of whom 35% sarcopenic) [57] for 5 years suggested that the combination of sarcopenia with a prognostic nutritional index called PNI (calculated from preoperative serum albumin concentration and peripheral blood whole lymphocyte count), could better predict the postoperative prognostic outcomes of HCC patients following hepatectomy than sarcopenia or PNI alone.

As for recurrence, only one study [47] evaluated the impact of sarcopenia on this outcome, in particular concerning the liver surface nodularity (LSN). The latter measures irregularities on the surface of the liver, is associated with fibrosis and portal hypertension [58,59], and has been shown to predict severe postoperative complications following liver resection for HCC [60]. This study recruited 110 patients with metabolic HCC and highlighted how LSN and sarcopenia are independent predictors of severe postoperative complications and recurrence following liver resection for HCC in patients with metabolic syndrome [47].

## 4. Impact of Sarcopenia on Locoregional Treatment Outcomes

Locoregional therapies encompass various approaches, including local ablation techniques such as radiofrequency ablation (RFA) and microwave ablation (MWA), TACE, and strategies involving internal radiation (TARE) or external stereotactic body radiotherapy (SBRT). The ablative therapies are recommended for individuals in the early stages of HCC (stages 0–A) and for patients for whom tumor resection is not a viable option [9]. TACE is generally suggested for patients at the intermediate stage (stage B) when nodules are well defined and portal flow is preserved. Alternatively, when early-stage HCC patients are not amenable to resection, liver transplant, or ablation, TACE is considered a viable treatment option [9]. Regarding radiation therapies, their role is less clearly defined by international guidelines. BCLC guidelines propose TARE for early-stage HCC patients (stages 0–A) with a single lesion ≤ 8 cm, while the role of SBRT requires further evidence [9]. A recent meta-analysis suggested that TARE is associated with a prolonged time to progression compared to TACE, without a significant difference in terms of OS [10].

Among thirty-one studies that aimed at evaluating the impact of sarcopenia on locoregional therapies, 19% (6 studies) focused on ablation, 58% (18 studies) on TACE, and 23% (7 studies) on TARE and SBRT. Also in this case, the majority of studies were retrospective, with only four being prospective.

Appendix A summarizes the general characteristics of studies evaluating the impact of sarcopenia on locoregional treatment outcomes.

### 4.1. Impact of Sarcopenia on Ablation Outcomes

Both RFA and MWA induce irreversible thermal damage to tumor mass using percutaneous probe insertion or laparoscopic access [61], and may lead to complications such as bleeding and hematoma development (less than 3% of patients), as well as fever and pain (5% of patients) [62].

Data on the role of sarcopenia on local ablation therapies remain inconclusive, primarily due to a limited number of studies specifically designed to assess the effect of sarcopenia on ablation therapies. Nevertheless, the presence of sarcopenia, primarily defined as a reduction in SMI or PMI at the third lumbar vertebra level on CT scan, appears to reduce OS following local ablation treatment, even if a smaller study on 56 HCC patients suggested a non-significant trend toward higher HCC recurrence in sarcopenic patients compared to non-sarcopenic ones [63].

This negative effect remains consistent across various HCC etiologies [64], even though in one prospective study involving patients with HCC due to HCV infection, sarcopenia was associated with lower OS after RFA at two-year follow-up, with nearly 8-fold higher risk of death [65], only in patients with successful viral eradication or without a history of HCV infection, but not in those with active HCV infection, probably due to the high percentage of sarcopenic and elderly patients in the HCV-active group [65].

Similarly, sarcopenia confirms its role in both sexes, as it has been identified as an independent factor associated with poor survival after local ablation in both males and females. However, when considering SMI as a continuous variable, it is associated with lower OS only in males [64].

On the contrary, it has not been demonstrated that there is any significant impact of sarcopenia on HCC recurrence [66,67].

Equally to resection, patients with sarcopenic obesity exhibit lower OS than overweight/obese patients without sarcopenia and sarcopenic patients with normal weight [66], along with a higher rate of HCC recurrence after treatment [68] even after ablation procedures. Furthermore, sarcopenic patients showed a higher percentage of major complications and a 90-day treatment-related mortality compared to patients without sarcopenia [66].

### 4.2. Impact of Sarcopenia on TACE Outcomes

TACE involves injecting a chemotherapeutic agent, usually an anthracycline or cisplatin, along with embolic material such as Lipiodol (cTACE) or non-resorbable embolic microspheres (DEB-TACE) into the vessels supplying the tumor mass [69], providing both a cytotoxic and an ischemic effect [70]. Both cTACE and DEB-TACE exhibited similar efficacy in terms of tumor response and OS. However, DEB-TACE carried a higher risk of adverse events, including hepatic and biliary damage, but it is associated with lower post-procedural pain [71].

Given the broad range of eligible patients for TACE [9], reported OS varied widely, ranging from 5 to 25 months, according to tumor size, lesions number, liver function, portal hypertension, and comorbidities [72]. Along with these conventional risk factors, sarcopenia, evaluated both as SMI or PMI, emerged as a notable risk factor for poor prognosis in HCC patients who undergo TACE [73,74,75,76,77,78]. This effect is particularly pronounced in male patients with lower BMI and older age [79], and it seems independent of tumor burden [80].

In studies that examined not only baseline sarcopenia but also changes in skeletal muscle and fat masses over the follow-up period post-TACE, a subsequent decrease in skeletal muscle mass [77,81] and concurrent reduction in subcutaneous fat with increased visceral adipose tissue negatively impact the OS after treatment [82]. Moreover, in patients undergoing multiple TACE procedures due to incomplete tumor response, progressive skeletal muscle mass reduction is associated with lower OS and decreased liver function reserve [83]. Conversely, regular exercise training supervised by a physical therapist leading to increased skeletal muscle mass after TACE results in higher survival rates [84,85]. However, these studies did not identify an independent correlation between changes in SMI and survival rates, nor did they find a correlation with baseline sarcopenia. Conversely, these two studies demonstrated that physical exercise, especially stretching, strength, balance, and endurance training, increased SMI in patients undergoing TACE [85], leading to prolonged survival in trained patients compared to inactive ones (529 versus 369 days, *p* = 0.03) [84].

As for resection and ablation, also for HCC patients treated with TACE, the combination of muscle depletion and increased visceral adiposity shown through CT scan (sarcopenic obesity) is a major adverse prognostic factor in terms of OS [86,87], mainly consequent to lower complete response rate after treatment and to impossibility to repeat the procedure because of frailty [86]. Indeed, data in the literature report how subjects with a progressive reduction in muscle mass after TACE require a higher number of TACE procedures compared to patients without it [82]. Despite these findings, one study conducted in 2019 observed that lower PMI did not predict tumor response after TACE, even if a non-significant negative trend was observed [77].

Even in transarterial embolization (TAE) without the use of a chemotherapeutic agents, sarcopenia negatively influences treatment outcomes, leading to higher mortality rates without an increase in adverse events or post-procedural complications [88].

Finally, myosteatosis negatively affects OS and treatment response in HCC patients undergoing TACE, with a more significant negative impact in the Asiatic population [89]. However, an Italian study did not observe a significant impact of myosteatosis on OS in HCC patients undergoing TAE [90].

### 4.3. Impact of Sarcopenia on Radiotherapy Outcomes

Another locoregional treatment has emerged in the context of HCC management, focusing on radiotherapy [9]. Radiotherapy options include TARE and SBRT. In TARE, Yttrium90 (Y90), a beta emitter radioisotope, is intravascularly administered with microspheres to the tumor mass, resulting in reduced systemic irradiation [91]. Y90 dose is tailored based on tumor size and hepatic volumetry [92]. SBRT employs a high dose of external radiation, typically delivered in 3–5 fractions to minimize damage to the remaining liver parenchyma [93]. While SBRT appears effective in patients with small HCC, to date its defined role is limited to unresectable cases and when other locoregional therapies are not feasible [9].

Sarcopenia seems to be an independent factor related to lower OS after radiotherapy in HCC patients, encompassing both TARE and SBRT [94,95]. As seen in HCC patients treated with TACE, and also in patients treated with radiotherapy, both baseline sarcopenia and the development [95] or worsening of sarcopenia after treatment [94] were associated with poor outcomes [94,95]. In SBRT-treated patients, a reduction in muscle mass during therapy, possibly attributed to the impact of radiation on muscle mass and associated appetite loss, was identified as a strong predictor of lower OS [94].

The detrimental correlation between sarcopenia and increased mortality after TARE was observed also using MRI for skeletal muscle mass evaluation [96,97], particularly in BCLC B stage [97]. On MRI, the paraspinal muscle area at the level of the superior mesenteric artery origin is a validated marker of sarcopenia, facilitating the evaluation of free fat muscle mass and the discrimination between sarcopenia and myosteatosis [14].

In 2023, a study introduced radiomics-based body composition analysis in HCC patients treated with sorafenib plus SIRT, revealing that alterations in skeletal muscle mass and adipose tissue quality were associated with lower OS [98].

The reduction in muscle mass after TARE is also linked to a higher HCC recurrence rate [95,99]. In fact, both male and female patients with a progression of neoplastic disease showed a further reduction in muscle mass before and after treatment, confirmed at three months. Conversely, patients with a complete response or stable disease did not exhibit significant changes in muscle mass [99].

Contrary to these findings, a retrospective study conducted in 2018 on carbon-ion radiotherapy did not show an impact of sarcopenia on both OS and PFS [100]. Carbon-ion radiotherapy is an external radiotherapy used in several solid tumors, including HCC, that uses carbon particles loaded with energy that can be directed with more precision to the tumor mass [101]. The efficacy of carbon-ion radiotherapy in both sarcopenic and non-sarcopenic patients is probably due to its lower invasiveness and toxicity [100].

## 5. Impact of Sarcopenia on Systemic Treatment Outcomes

According to current guidelines, patients with diffuse, infiltrative, and bilobar liver involvement (BCLC stage B) or with portal and/or extrahepatic spread of HCC (BCLC stage C) are eligible for systemic therapy [9]. Historically, TKIs were the sole treatment option for advanced HCC [102]. Since 2020, the combination therapy with the immune checkpoint inhibitor (ICI) atezolizumab and the anti-angiogenic bevacizumab (ATZ/BEV) has become available. Notably, due to superior OS of patients treated with ATZ/BEV compared to sorafenib [11], ATZ/BEV currently stands as the first-line systemic treatment [9]. Among TKIs, sorafenib, regorafenib, lenvatinib (LEN), and cabozantinib are approved for HCC treatment as second-line or first-line in patients with contraindications to the use of ICI [9].

Given the relatively recent introduction of ICIs, no prospective studies are available and most of the studies have focused on the impact of sarcopenia on TKI outcomes.

Among prognostic factors that affect OS after therapy with TKI, such as lower albumin, increased alfafetoprotein, multiple nodules with bilobar involvement, and portal vein invasion, sarcopenia appeared to be a major determinant of poor survival, especially in HCC patients with few negative prognostic factors [103].

In patients treated with sorafenib [104,105,106] and LEN [107,108], baseline sarcopenia, defined as a reduction in muscle mass on CT scan, emerged as an independent predictor of OS across age and BCLC stage. Additionally, sarcopenia correlated with poor performance status and reduced liver function tests [78,104,105,107,108].

The association between sarcopenia and poor OS remained consistent, whether assessed through SMI or psoas muscle area [109], although psoas muscle thickness reduction during therapy with sorafenib appeared more linked to lower OS and PFS than its baseline measurement [110].

Interestingly, the prognostic effect of sarcopenia with sorafenib and LEN treatment appeared more pronounced in males, possibly due to the higher prevalence of HCC in this gender [78]. However, a European study conducted on a Caucasian-Mediterranean cohort revealed a higher prevalence of sarcopenia in females treated with sorafenib [106].

In patients treated with LEN, sarcopenia correlated with lower PFS, albeit without reaching significance [107].

Moreover, sarcopenia was related to a higher prevalence of adverse events, such as severe diarrhea, in patients treated with sorafenib [106], possibly due to higher serum levels of sorafenib for reduced body mass [111] or to appetite loss [108,112], and encephalopathy and ascites in those treated with LEN [112].

Not only baseline sarcopenia is an independent negative prognostic factor in HCC patients treated with TKI, but also the further reduction in skeletal muscle mass during treatment with sorafenib or LEN seems to have an impact [113,114,115], independently of drug used, type of adverse events, tumor stage, and liver functional reserve [113]. In fact, for patients progressing after sorafenib, sarcopenia, along with older age, female sex, and low BMI, was associated with a poor prognosis [116], whereas among patients with baseline sarcopenia, those who improved muscle mass during sorafenib treatment exhibited longer post-progression survival compared to those with further muscle depletion [116].

Finally, only one study assessed the relationship between muscle strength, evaluated through the hand-grip test, and the OS in patients treated with LEN [117]. A low muscle strength was independently associated with poor prognosis and early treatment discontinuation due to adverse events. The negative impact of low muscle strength seemed to be stronger when compared to the impact of low muscle mass [117].

While data regarding the impact of sarcopenia on TKI-treated patients were more concordant, emerging data on ICI are more inconclusive and conducted only in Asian patients.

Studies evaluating the impact of sarcopenia in patients treated with ATZ/BEV, combining anti-programed death ligand-1 (PDL1) antibody and anti-vascular endothelial growth factor (VEGF) antibody [11], showed that reduced SMI during treatment correlated with lower OS and PFS, primarily influenced by age and not by gender and etiology of HCC [118]. Similarly, a reduction in nutritional assessment, as measured by tools like the Geriatric Nutritional Risk Index that encompasses albumin, patients body weight, and optimal body weight, led to reduced OS and PFS in patients treated with ATZ/BEV [119].

Conversely, the comparison between the prognostic effect of sarcopenia in HCC patients treated with LEN and ATZ/BEV showed that the presence of sarcopenia, evaluated by CT scan, negatively affected the prognosis of patients treated with LEN but not with ATZ/BEV [120]. Moreover, in LEN-treated patients, sarcopenia seemed to reduce OS but not PFS [120].

One study incorporating grip strength and SMI at BIA in ATZ/BEV-treated patients [121] showed higher adverse events, particularly anorexia and diarrhea, in sarcopenic patients. Sarcopenia was also independently associated with reduced OS and worsening of liver function, leading to early therapy discontinuation [121].

Various anti-programed cell death 1 (PD1), such as pembrolizumab, nivolumab, sintilimab, and camrelizumab are currently under investigation for the treatment of HCC [122]. In a Chinese study involving TKIs (LEN or sorafenib) plus a combination of ICIs, patients with sarcopenia at baseline experienced a reduced OS and an increased disease progression compared to patients without it in all regimen choices [123]. A European study conducted on patients treated with various anti-PDL1 regimens confirmed that sarcopenia negatively affected OS, especially in patients with higher systemic inflammation activation [124]. On the other hand, in camrelizumab-treated patients, sarcopenia was related to lower PFS but not to a lower OS, even if a trend toward lower OS in sarcopenic individuals was observed [125]. The presence of sarcopenia was related to higher inflammatory markers and lower levels of albumin, another marker of both malnutrition and reduced liver function [125].

Moreover, in patients treated with nivolumab with or without associated radiotherapy, sarcopenia reduced OS in patients treated with nivolumab, whereas the combination of nivolumab plus radiotherapy seemed to improve the OS in sarcopenic patients [126].

Finally, a study conducted in HCC patients treated with ICIs showed that sarcopenia and lower subcutaneous adipose tissue index, a predictor of cancer mortality [127], were both related to low OS, leading to the development of a body composition-based nomogram to predict survival in patients treated with ICIs [128].

Lastly, both sarcopenia and myosteatosis appeared to be related to poor prognosis in patients treated with a combination of anti-PDL1 and anti-cytotoxic T-lymphocyte-associated protein 4 (CTLA-4), whereas myosteatosis was also related to lower disease control rates and lower PFS [129].

Appendix A summarizes the general characteristics of studies evaluating the impact of sarcopenia on systemic treatment outcomes.

## 6. Discussion

According to a recent meta-analysis of 25 studies, mainly conducted in Asia and encompassing 5522 patients with HCC, sarcopenia is highly prevalent among these patients. Its occurrence, however, varies across studies (ranging from 11 to 85%, with a cumulative incidence of 38.5%), probably because of differences in the criteria used to define it [130]. In fact, several studies considered SMI as a marker of skeletal muscle mass, while others considered the use of PMI. Despite these variations, the presence of sarcopenia independently correlated with lower OS and PFS after surgical, locoregional, and systemic therapies including TKIs and ICIs. Notably, while sarcopenia is associated with a more severe liver disease and a lower patient performance status [131,132], its negative impact on OS and PFS remained independent of the underlying liver function or the etiology of HCC, despite a lower prevalence of sarcopenia in patients with metabolic liver disease [80].

Data in the literature report also an association between sarcopenia and a higher incidence of treatment adverse events, particularly in patients receiving surgery and TKIs [106,108,112].

A schematic effect of the impact of sarcopenia on both surgical resections, locoregional treatments, and systemic therapies is represented in Figure 1.

The relationship between sarcopenia and HCC is intricate and not fully elucidated. An upregulation in pro-inflammatory cytokines pathways and a dysregulation in lipid metabolism with a reduction in sterol lipids and fatty acyls are observed in HCC patients with sarcopenia [133]. As a consequence, lipid accumulation and peroxidation and the amplification of the production of reactive oxygen species (ROS) into the hepatocytes foster cancer initiation and progression [134]. Moreover, the loss of skeletal mass contributes to the development of insulin resistance, subsequently increasing insulin-like growth factor-1 (IGF-1) levels. IGF-1, acting through the mammalian target of rapamycin (mTOR) pathway, further upregulates hepatocytes proliferation, increasing the risk of HCC development [17].

Sarcopenia is also associated with larger tumor size and greater number of lesions at the time of diagnosis, impacting both treatment allocation and outcomes [79]. The underlying mechanisms linking sarcopenia to OS after treatment in HCC patients remain not completely understood. However, the combination of low muscle mass and function, malnutrition, and lower BMI probably reduce the capacity of patients to handle cancer therapy. Furthermore, patients eligible for systemic therapy had a higher prevalence of sarcopenia compared to those treated with surgical or locoregional treatments, with prevalence ranging from 15% to 65% depending on geographical region and on the criteria used to define sarcopenia. The high prevalence of sarcopenia in patients with advanced HCC likely arises from the strict correlation between the progressive decline in skeletal muscle mass and the severity of malignancy [135]. The higher prevalence of sarcopenia among patients treated with systemic therapy is probably linked to its negative impact on treatment outcomes, as observed after surgery and locoregional therapies.

While both sarcopenia and HCC are more common in males [136,137], possibly consequent to the protective role of estrogen on muscle mass [138] and carcinogenesis in females [139], only a limited number of studies have specifically considered gender in evaluating the impact of sarcopenia on HCC treatment outcomes. As expected, the negative impact of sarcopenia on OS after treatment is more evident in males [31,64,78,79], even if a study conducted on a Caucasian-Mediterranean cohort revealed a higher prevalence of sarcopenia in female candidates undergoing systemic therapy [106]. This divergence might be explained by the role of ethnicity in sarcopenia development and body composition [140]. Most studies evaluating the impact of sarcopenia on HCC treatment outcomes were conducted in Asian populations and body composition and cut-offs used to define sarcopenia are different across geographic areas [141].

While most studies in this review demonstrated the detrimental effects of baseline sarcopenia on HCC treatment outcomes, some others examining changes in skeletal muscle mass before and after treatments indicated that eventually alterations in muscle mass during treatment, rather than baseline sarcopenia, are the primary determinants of reduced OS.

This review has several limitations. Firstly, the majority of the studies included were retrospective, preventing the establishment of a causal relationship between sarcopenia and poor prognosis. Consequently, the lack of higher-level evidence affects these studies.

Moreover, only few studies considered muscle strength in addition to muscle mass to define sarcopenia. Indeed, current guidelines on sarcopenia recommend evaluating both muscle mass and strength to define sarcopenia [14], and studies incorporating both of them showed that low muscle strength, rather than muscle mass, was associated with a poor prognosis in HCC treatment [117]. Additionally, myosteatosis showed an impact on HCC treatments outcomes [44,55,56], especially after surgery [44,55,56], whereas its role after locoregional treatment is not well defined [89,90] and further studies are needed to specifically address this issue. Furthermore, while the correlation between sarcopenia and low OS after HCC treatment is consistently observed in all studies reviewed, the effect of sarcopenia on other treatment outcomes, such as early and late HCC recurrence and the extrahepatic spread of tumor, remained under investigated. Finally, sarcopenia has been identified also in the early stages of liver disease and HCC can arise also in non-cirrhotic livers [16,142]; however, the relationship between sarcopenia and HCC treatment outcomes has been evaluated only in cirrhotic patients.

The strength of the review is that it involves a comprehensive evaluation of the impact of sarcopenia and body composition among all validated treatments for HCC, considering both OS and disease progression.

Further prospective studies are required to establish the causal relationship between sarcopenia and HCC treatment outcomes, particularly emphasizing changes in skeletal muscle mass before and after treatment. If the negative impact of sarcopenia on HCC treatment outcomes will be confirmed in a prospective and randomized controlled trial, a treatment algorithm for HCC patients that accounts for the presence of sarcopenia should be considered to tailor a more personalized therapeutic approach for each patient. Moreover, there is an urgent need for clinical trials aimed at evaluating the impact of both physical activity and diet to enhance muscle mass and function during HCC treatment, as well as their impact on OS, PFS, and adverse events.

## 7. Conclusions

In conclusion, despite the retrospective nature of the majority of studies included in the review, given the unfavorable prognosis observed in patients with HCC and low muscle mass both before initiating treatment and during the course of treatment (including surgical, locoregional, and systemic treatments), it is crucial to assess the presence of sarcopenia before treatment initiation. Despite there being a lack of clinical trials assessing the impact of physical activity and nutritional intervention on restoring muscle mass and their effects during HCC treatment, the evaluation of sarcopenia is essential for predicting patients’ responses and tailoring a more personalized treatment approach for those with sarcopenia. Furthermore, initiation of a treatment aimed to restore muscle mass, preventing its further deterioration over time [84], has been demonstrated to be essential to improve OS in these patients. However, several issues are still unanswered, so the relationship between sarcopenia and treatment outcomes require further investigation in order to delineate a personalized approach for patients with HCC.

## Figures and Tables

**Figure 1 cancers-16-02315-f001:**
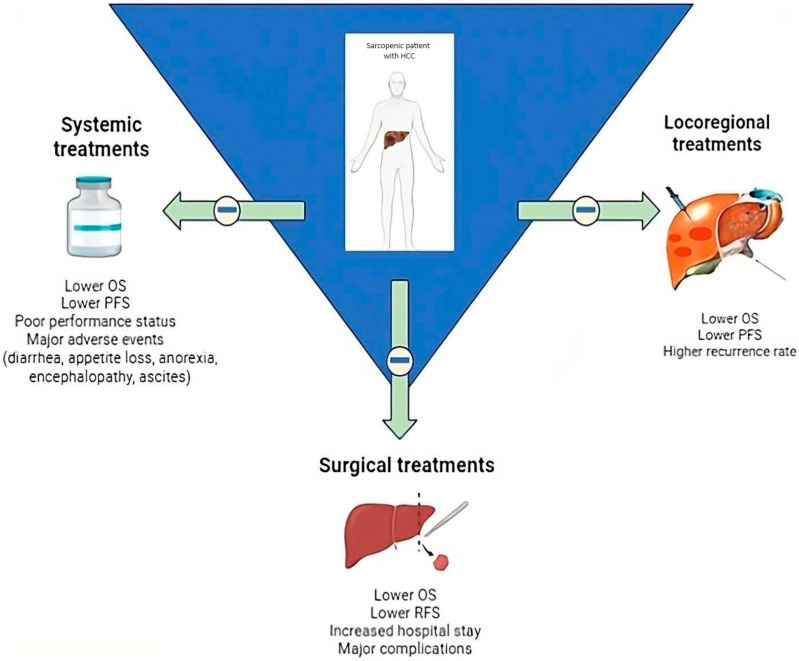
The impact of sarcopenia, expressed as a reduction in skeletal muscle mass, in hepatocellular carcinoma (HCC) treatments outcomes. The presence of sarcopenia is associated with unfavorable prognosis following surgical resection, locoregional treatments, and systemic therapies. Furthermore, the occurrence of sarcopenia is related to higher rates of adverse events in both surgical and systemic treatment options. In the context of locoregional therapies like radiofrequency ablation (RFA) and transarterial chemoembolization (TACE), baseline sarcopenia is associated with an increased risk of HCC recurrence after treatment.

**Table 1 cancers-16-02315-t001:** Parameters and their definitions used to describe sarcopenia and myosteatosis across included studies.

Parameter	Abbreviation	Definition	Type of Evaluation
CT scan			
Skeletal muscle index	SMI	Muscle mass at third lumbar vertebra/height^2^	Sarcopenia
Psoas muscle index	PMI	Left + right psoas major muscle area/height^2^	Sarcopenia
Psoas area and volume		Measure of right and left psoas at the level of third lumbar vertebra (area) or measure of right and left entire psoas muscle (volume)	Sarcopenia
Skeletal muscle mass-to-visceral fat area ratio	SVR	Skeletal muscle mass at third lumbar vertebra/area of tissue with densities ranging from −190 to −30 HU from a single axial slice at the umbilicus level	Sarcopenia
Cross-sectional area of paraspinal muscles	CSA	Area of paraspinal muscles at the third lumbar vertebra level	Sarcopenia
Skeletal muscle mass radiation attenuation	SMRA	Muscle with attenuation between −29 and +150 HU at the third lumbar vertebra level	Myosteatosis
Skeletal muscle density	SMD	Areas of the abdominal wall and back muscles with attenuation between −29 and +150 HU	Myosteatosis
Intramuscular adipose tissue content	IMAC	Attenuation values of the multifidus muscles/attenuation values of the subcutaneous fat	Myosteatosis
MRI			
Paraspinal muscle mass		Left and right superficial and deep paraspinal muscles at the superior mesenteric artery origin level	Sarcopenia
Fat-free skeletal muscle area	FFMA	Subtraction of low-signal-intensity pixels from the paraspinal muscle area	Myosteatosis
BIA			
Skeletal muscle index	SMI	Skeletal muscle mass obtained by BIA/height^2^	Sarcopenia
TEST			
Grip strength		Average of four trials, with two trials conducted for each hand	Muscle strength
Five-time chair standing test		Time to stand up from the chair five times	Muscle strength
Gait speed		Speed for walking on 8 m in a straight line	Muscle strength

BIA: bioimpedance analysis; CT: computed tomography; MRI: magnetic resonance imaging.

## Data Availability

Not applicable.

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
