# Peer review of "The Negative Impact of Sarcopenia on Hepatocellular Carcinoma Treatment Outcomes"

_cancers, 2024, doi:10.3390/cancers16132315_

Round 1

Reviewer 1 Report

Comments and Suggestions for Authors

This is a thorough and comprehensive review of the impact of sarcopenia on the outcome of HCC

My comments:

A small table/text box is wanted for the various definitions of sarcopenia

Research strategy is well described

Chapters 3 and 4 are fine

Refs 73 and 74 should be explained in more detail since these interventional studies are pivotal for the review

Figure 1 is fine. Explain better ‘major adverse events’

In discussion please suggest an outline for future studies

Comments on the Quality of English Language

Could be improved

Author Response

This is a thorough and comprehensive review of the impact of sarcopenia on the outcome of HCC. My comments:

We thank the Reviewer for her/his insightful revision of the manuscript and for the suggestions.

Please see below the point-to-point answers to the Reviewers. Changes in the manuscript are marked up using the “Track Changes” (which are in red) and highlighted in yellow.

Point 1: A small table/text box is wanted for the various definitions of sarcopenia.

Response 1: We appreciate the Reviewer’s comment and, according to his/her suggestion, we have included a table summarizing the parameters used to describe sarcopenia and myosteatosis across the studies included in the review.

Point 2: Research strategy is well described.

Response 2: We thank the Reviewer for his/her acknowledgment regarding the “Research strategy and study selection” section.

Point 3: Chapters 3 and 4 are fine.

Response 3: We thank the Reviewer for his/her feedback on Chapters 3 and 4.

Point 4: Refs 73 and 74 should be explained in more detail since these interventional studies are pivotal for the review.

Response 4: We completely agree with the reviewer and appreciate the suggestion. We have now provided a more detailed explanation of the type of exercise performed in Refs 73 and 74, as well as the benefits in terms of overall survival.

Point 5: Figure 1 is fine. Explain better ‘major adverse events’.

Response 5: We thank the Reviewer for his/her observation. We have clarified the major adverse events observed during systemic treatments based on the presence of sarcopenia.

Point 6: In discussion please suggest an outline for future studies

Response 6: We thank the Reviewer for his/her suggestion. We have included possible future studies in the Discussion session.

Reviewer 2 Report

Comments and Suggestions for Authors

Thank you for the opportunity to review this interesting manuscript. Drs Cespiati et al have undertaken an ambitious and timely narrative review of sarcopenia and HCC outcomes. The paper synthesises many studies, mostly retrospective, which document varied associations of sarcopenia with different HCC outcomes. I do have some concerns that the representations made could overstate the field of play, and imply more causation than be attributed with the current evidence. I think the manuscript could be more helpful if it applied a more critical eye to the current level and quality of evidence in the included studies. I am also concerned that some of the findings in the supplementary tables have not been included in the summary statements in the results, as detailed below.

Specifically:

-              The title “detrimental effect” implies causation and could be reconsidered

-              Comments in the abstract and conclusions suggesting the need to ‘address sarcopenia’ and ‘tailor treatment’ based on baseline sarcopenia measurements do not seem backed up by the literature presented which includes very few interventional studies

-              The Supplementary tables, and largely the results, provide some important details about study factors, but do not include information on the study type (i.e. are they all retrospective cohort studies) nor whether the associations seen are on univariable or multivariable analysis.

-              For ease of reference between the text and the supplementary tables, it would be ideal if the supplementary tables can be cross-linked to their appropriate reference #.

-              The methods suggest a systematic approach to the literature review, but provide incomplete documentation of this process. Ideally this would include some more detail on search terms, and the specific inclusion and exclusion criteria used to identify and screen out articles.

-              Ideally throughout, the results should also highlight negative findings and discordant findings between studies, and ensure consistency with the reported findings in the text and those presented in the Supplementary tables.

o   For instance, in the transplantation section, Valero [33] is used to highlight an increased report of adverse events with sarcopenia, but is also documented in Suppl Table 1 to show no impact of sarcopenia on overall survival. This is not mentioned earlier in the results section about Overall Survival.

o   In the surgical resection section, 3.2, the opening statement paragraph from Line 172 suggests that all studies on overall survival showed an association. However, discordant results are seen in Valero and Harimoto (presented in Supp Table 1), and Beumer only saw this in a subgroup of paper. Similarly, Itoh did not report a decrease in RFS.

o   The paragraph from line 222 states only one study considers recurrence. But previously many studies considering recurrence free survival have been mentioned? I’m unclear what distinct entity is being addressed in this paragraph.

o   In Section 4.1, deals with recurrence in both lines 260 and 272 separately, referencing different studies. These report no impact on HCC recurrence, although in Supp Table 2, Jaruvongvanich & Kamachi both are reported to show an association with HCC recurrence. Which is true?

o   Why does the MA of studies show no association of sarcopenia and survival post TARE, but the studies presented in section 4.3 only report studies that show an association between sarcopenia and mortality post TARE?

-              In addition, throughout the results (ideally including the presented Figure), the synthesis of studies should highlight at a minimum the nature of the studies (i.e. level of evidence) being used to draw conclusions.

-              There appear to be 1-2 studies trying to intervene on sarcopenia with LORT that are presented. These are particularly important. Reading reference 73, while cancer rehabilitation both showed an improvement in sarcopenia and an increase in survival, survival was not increased among patients with increased skeletal muscle area, nor was baseline sarcopenia a significant factor. This discordant finding should be highlighted.

-              The discussion reports at 35% cumulative incidence of sarcopenia among patients with HCC – it is unclear where this statistic has been drawn from, in what population, or over what time frame.

-              Lines 522-524 again overstate that all studies found an association of sarcopenia with overall survival being seen in every study examined, which is discordant with the data presented in the Supplementary Tables.

-              The discussion does well to point out weaknesses including the nature of the studies, and mentions that there are many possible confounding factors for the impact of sarcopenia on survival including its association with tumour stage. The paper calls for more detailed studies on HCC recurrence and spread being associated with sarcopenia. Perhaps it would be more fruitful to comment on areas of study that might impact practice more directly – for instance, how can sarcopenia be used to adjust treatments, or be incorporated into prognostic algorithms? Are there interventions that can address sarcopenia in the setting of HCC? These studies require more investment, but would be more likely to change practice than more retrospective data.

Author Response

Thank you for the opportunity to review this interesting manuscript. Drs Cespiati et al have undertaken an ambitious and timely narrative review of sarcopenia and HCC outcomes. The paper synthesises many studies, mostly retrospective, which document varied associations of sarcopenia with different HCC outcomes. I do have some concerns that the representations made could overstate the field of play, and imply more causation than be attributed with the current evidence. I think the manuscript could be more helpful if it applied a more critical eye to the current level and quality of evidence in the included studies. I am also concerned that some of the findings in the supplementary tables have not been included in the summary statements in the results, as detailed below.

We thank the Reviewer for her/his thoughtful and thorough revision of the manuscript and for the relevant suggestions.

Please see below the point-to-point answers to the Reviewers. Changes in the manuscript are marked up using the “Track Changes” (which are in red) and highlighted in yellow.

Point 1: The title “detrimental effect” implies causation and could be reconsidered.

Response 1: We appreciate the Reviewer’s suggestion. In response, we have changed the review title by replacing “detrimental effect” with “negative impact”.

Point 2: Comments in the abstract and conclusions suggesting the need to ‘address sarcopenia’ and ‘tailor treatment’ based on baseline sarcopenia measurements do not seem backed up by the literature presented which includes very few interventional studies.

Response 2: We thank the Reviewer for pointing this out. There are very few studies evaluating the impact of physical activity and nutritional assessment aimed at restoring muscle mass in HCC treatment outcomes. We have emphasized this limitation in both the abstract and conclusion sections.

Point 3: The Supplementary tables, and largely the results, provide some important details about study factors, but do not include information on the study type (i.e. are they all retrospective cohort studies) nor whether the associations seen are on univariable or multivariable analysis.

Response 3: We thank the Reviewer for this observation. We now categorized the articles in Supplementary Tables 1, 2, and 3 based on whether the studies are prospective or retrospective. Moreover, we have clarified in the manuscript that the majority of the included studies are retrospective. Regarding the results, all data reported in the review were derived from multivariate analysis, which is now indicated in Supplementary Tables 1, 2, and 3.

Point 4: For ease of reference between the text and the supplementary tables, it would be ideal if the supplementary tables can be cross-linked to their appropriate reference #.

Response 4: We thank the Reviewer for this comment. We have now implemented the Supplementary Tables, adding the references for each article.

Point 5: The methods suggest a systematic approach to the literature review, but provide incomplete documentation of this process. Ideally this would include some more detail on search terms, and the specific inclusion and exclusion criteria used to identify and screen out articles.

Response 5: We thank the Reviewer for his/her suggestion. We have better described the inclusion and exclusion criteria, as well as the search terms used in our narrative review.

Point 6: Ideally throughout, the results should also highlight negative findings and discordant findings between studies, and ensure consistency with the reported findings in the text and those presented in the Supplementary tables.

Response 6: We appreciate the Reviewer’s remarks regarding the inclusion of negative and discordant results of the studies under review and proceeded to integrate them through the result section of our paper. In particular:

6.1 For instance, in the transplantation section, Valero [33] is used to highlight an increased report of adverse events with sarcopenia, but is also documented in Suppl Table 1 to show no impact of sarcopenia on overall survival. This is not mentioned earlier in the results section about Overall Survival.

Response 6.1: We thank the Reviewer for highlighting this oversight. The non-significant association of sarcopenia with OS found by Valero and colleagues has now been noted in the results regarding OS in the transplant section.

6.2 In the surgical resection section, 3.2, the opening statement paragraph from Line 172 suggests that all studies on overall survival showed an association. However, discordant results are seen in Valero and Harimoto (presented in Supp Table 1), and Beumer only saw this in a subgroup of paper. Similarly, Itoh did not report a decrease in RFS.

Response 6.2: We appreciate the Reviewer’s observation. Regarding the study by Valero et al, although there was no association with long-term outcomes such as OS, their cohort included patients undergoing resection for both HCC and cholangiocarcinoma and, while post-operative complications were significantly higher in sarcopenic patients, the lack of correlation with long-term outcomes is likely due to the differing prognoses of these two diseases, as noted by the authors. According to the Reviewer's observation, we have expanded our discussion to include the observations regarding the studies of Beumer, Harimoto, and Itoh.

6.3 The paragraph from line 222 states only one study considers recurrence. But previously many studies considering recurrence free survival have been mentioned? I’m unclear what distinct entity is being addressed in this paragraph.

Response 6.3: We thank the Reviewer for pointing this out. The mentioned paragraph indeed discusses studies regarding correlation with myosteatosis rather than sarcopenia. We have revised the text to better highlight this aspect.

6.4 In Section 4.1, deals with recurrence in both lines 260 and 272 separately, referencing different studies. These report no impact on HCC recurrence, although in Supp Table 2, Jaruvongvanich & Kamachi both are reported to show an association with HCC recurrence. Which is true?

Response 6.4: We appreciate the Reviewer’s remarks. The results from Jaruvongvanich’s study were inaccurately reported in Supplementary Table 2 since this study reported no significant association between RFS and sarcopenia, but only a trend (P 0.052), probably due to the reduced sample size. We have corrected Supplementary Table 2 accordingly. On the other hand, Kamachi et al. reported a significant association, as correctly noted, and we have corrected the corresponding sentences.

6.5 Why does the MA of studies show no association of sarcopenia and survival post TARE, but the studies presented in section 4.3 only report studies that show an association between sarcopenia and mortality post TARE?

Response 6.5: Regarding radiotherapy, 6 out of 7 of the studies showed an association between sarcopenia and reduced OS or PFS, as we reported in our result section regarding this treatment and in the Supplementary Table 2.

Point 7: In addition, throughout the results (ideally including the presented Figure), the synthesis of studies should highlight at a minimum the nature of the studies (i.e. level of evidence) being used to draw conclusions.

Response 7: We appreciate the Reviewer’s observation. We have highlighted the characteristics of the studies in both the Results and the Supplementary Tables. Additionally, we have emphasized the lack of high-level evidence in the Discussion.

Point 8: There appear to be 1-2 studies trying to intervene on sarcopenia with LORT that are presented. These are particularly important. Reading reference 73, while cancer rehabilitation both showed an improvement in sarcopenia and an increase in survival, survival was not increased among patients with increased skeletal muscle area, nor was baseline sarcopenia a significant factor. This discordant finding should be highlighted.

Response 8: We thank the Reviewer for his/her observation and we have added this discordant finding in the result section regarding physical rehabilitation in TACE.

Point 9: The discussion reports at 35% cumulative incidence of sarcopenia among patients with HCC – it is unclear where this statistic has been drawn from, in what population, or over what time frame.

Response 9: We thank the Reviewer for pointing this out. We have now clarified the characteristics of the meta-analysis used to determine the cumulative incidence of sarcopenia among patients with HCC.

Point 10: Lines 522-524 again overstate that all studies found an association of sarcopenia with overall survival being seen in every study examined, which is discordant with the data presented in the Supplementary Tables.

Response 10: We thank the Reviewer for bringing this oversight to our attention. We have reworded the concept accordinglyt.

Point 11: The discussion does well to point out weaknesses including the nature of the studies, and mentions that there are many possible confounding factors for the impact of sarcopenia on survival including its association with tumour stage. The paper calls for more detailed studies on HCC recurrence and spread being associated with sarcopenia. Perhaps it would be more fruitful to comment on areas of study that might impact practice more directly – for instance, how can sarcopenia be used to adjust treatments, or be incorporated into prognostic algorithms? Are there interventions that can address sarcopenia in the setting of HCC? These studies require more investment, but would be more likely to change practice than more retrospective data.

Response 11: We sincerely thank the Reviewer for these observations, and we completely agree. We have now emphasized the urgent need for prospective studies and randomized controlled trials to evaluate the causal relationship between sarcopenia and HCC treatment outcomes, as well as the impact of potential sarcopenia treatments on OS and PFS.